# Properties and Biodegradation of Poly(lactic Acid)/Thermoplastic Alginate Biocomposites Prepared via a Melt Blending Technique

**DOI:** 10.3390/polym17101338

**Published:** 2025-05-14

**Authors:** Yodthong Baimark, Kansiri Pakkethati, Prasong Srihanam

**Affiliations:** Biodegradable Polymers Research Unit, Department of Chemistry and Centre of Excellence for Innovation in Chemistry, Faculty of Science, Mahasarakham University, Mahasarakham 44150, Thailand; kansiri.p@msu.ac.th (K.P.); prasong.s@msu.ac.th (P.S.)

**Keywords:** poly(lactic acid), thermoplastic alginate, biocomposites, mechanical properties, biodegradation

## Abstract

In this work, poly(L-lactic acid)/thermoplastic alginate (PLA/TPA) biocomposites were prepared through a melt blending method. The TPA was initially prepared using glycerol as a plasticizer. The effects of TPA content on the interactions between blend components, thermal properties, phase morphology, mechanical properties, hydrophilicity, and biodegradation properties of biocomposites were systematically investigated. Fourier transform infrared (FTIR) spectroscopy analysis corroborated the interaction between the blend components. The addition of TPA enhanced the nucleating effect for PLA, as determined by differential scanning calorimetry (DSC). Scanning electron microscopy (SEM) revealed poor phase compatibility between the PLA and TPA phases. The thermal stability and mechanical properties of the biocomposites decreased with the addition of TPA, as demonstrated by thermogravimetric analysis (TGA) and tensile tests, respectively. The hydrophilicity and soil burial degradation rate of biocomposites increased significantly as the TPA content increased. These results indicated that PLA/TPA biocomposites degraded faster than pure PLA, making them suitable for single-use packaging, but this necessitates careful optimization of TPA content to balance mechanical properties and soil burial degradation rate for practical single-use applications.

## 1. Introduction

The use of plastic products derived from non-biodegradable plastics such as polyethylene (PE) and polypropylene (PP) results in pollution challenges associated with plastic waste, especially in single-use packaging applications [1,2,3,4,5]. This pollution negatively impacts ecosystems and leads to prolonged environmental degradation, given that these materials may require hundreds of years to degrade [4]. Consequently, there is a pressing demand for sustainable alternatives and improved waste management practices. This trend has led to significant advancements in the development of biodegradable plastics aimed at replacing non-biodegradable alternatives, with the objective of minimizing plastic waste, particularly through the utilization of bio-based plastics or bioplastics for a more sustainable approach to plastic consumption [5]. The objective of these initiatives is to reduce the ecological footprint of conventional plastics by developing substances that break down more readily in natural settings.

Among biodegradable bioplastics, poly(L-lactic acid) (PLA) has attracted significant interest due to its biocompatibility, excellent mechanical properties, and availability in the market [6,7,8]. Nonetheless, PLA continues to be costly in comparison to petroleum-derived commodity plastics like PE and PP, which restricts its broader utilization. Consequently, current investigations focus on lowering the production costs of PLA products, potentially increasing their competitiveness across a wider array of applications.

One significant limitation of PLA is its prolonged biodegradation time, particularly in landfill environments. This phenomenon raises environmental concerns, as PLA may endure longer than anticipated under these conditions. Consequently, investigations are underway to improve its soil burial degradation rate and minimize its environmental footprint. Researchers have composited PLA with cost-effective biopolymers like starch [9,10,11], chitosan [12], and alginate [13] to address these challenges and lower production costs. This innovative approach improves the environmental sustainability of PLA and creates new opportunities for producing biodegradable materials. By optimizing the formulation of these PLA-based biocomposites, the goal is to create products that are both economically viable and environmentally friendly.

Starch, as a biopolymer, has garnered significant attention due to its ability to be transformed into thermoplastic starch (TPS) through the incorporation of plasticizers such as glycerol. This process breaks the hydrogen bonds in starch, making it easier to create TPS products with conventional processing machines [14,15,16]. As a result, PLA/TPS biocomposites can be produced via the melt blending method, with TPS demonstrating a more economical profile compared to PLA and showing higher hydrophilicity than PLA. PLA/TPS biocomposites demonstrate a lower production cost and a faster biodegradation in comparison to PLA [15,17]. Research on mixing different biopolymers with PLA is gaining interest to create cheaper PLA-based products that biodegrade faster than those made only from PLA. This method improves the ecological advantages of PLA and creates new opportunities for creative uses in single-use packaging. By optimizing the blend ratios, the goal is to tailor the properties of these biocomposites to meet specific performance requirements while maintaining sustainability.

Alginate is a biopolymer sourced from brown algae, characterized by its non-toxic nature, cost-effectiveness, and widespread availability in the environment [18,19,20]. This material is a biodegradable biopolymer that has undergone extensive investigation and development for various applications across sectors such as biomedicals [21,22,23], wastewater treatment [24,25,26], and packaging [19,27], with a primary focus on utilizing alginate solutions to create alginate devices.

The conversion of alginate into thermoplastic alginate (TPA), which can be melt-molded using polyols as plasticizers, has been documented [27,28]. We expect that the process of melting TPA with PLA will lead to a decrease in production costs and enhance the biodegradation rate of PLA-based biocomposites. However, there are no documented studies regarding the preparation and analysis of the properties of PLA/TPA biocomposites. Investigating the synthesis of TPA and its blending with PLA was the aim of this study. The characteristics of the PLA/TPA biocomposites were studied in terms of their phase morphology, thermal properties, mechanical properties, hydrophilicity, and soil burial degradation.

## 2. Materials and Methods

### 2.1. Materials

PLA (3251D injection grade) was obtained from NatureWorks LLC (Waltham, MA, USA). It has a melt flow index (MFI) of about 30 g/10 min (190 °C/2.16 kg). Sodium alginate (SA) with a particle size ≤ 170 mesh and viscosity of 800 cps measured in 1% solution at 20 °C (Brookf. DV3T No. 62/20 rpm) was obtained from Chanjao Longevity Co., Ltd. (Bangkok, Thailand). Glycerol (QReC brand, 99.5%, AR grade) was purchased from Smart Science Co., Ltd. (Pathum Thani, Thailand).

### 2.2. Preparation of TPA

Sodium alginate (7 g) and glycerol (3 g) mixtures were kneaded and rolled until homogenous to form thermoplastic alginate (TPA) before being cut into pellets with scissors. The TPA pellet preparation process is shown in Figure 1.

### 2.3. Preparation of PLA/TPA Biocomposites

PLA was melt-blended with TPA using an internal mixer (Polylab OS System, HAAKE, Waltham, MA, USA) at 170 °C for 8 min with a rotor speed of 100 rpm. The PLA and TPA were dried in a convection oven at 80 °C for 6 h before melt blending. The biocomposites with TPA contents of 5 wt%, 10 wt%, 20 wt%, and 30 wt% were prepared and investigated. The biocomposites were dried in a convection oven at 80 °C for 6 h before film forming. The biocomposites were then hot-pressed into films using a compression molding machine (Auto CH, Carver, Wabash, IN, USA). A compression molding temperature of 175 °C, compression force of 10 MPa, and compression time of 3 min were used. The resulting film had a thickness of 0.2–0.3 mm. The pure PLA films were also prepared under the same conditions for comparison. The resulting films were kept in a desiccator for 24 h before characterization.

### 2.4. Characterization of TPA and PLA/TPA Biocomposites

An attenuated total reflectance–Fourier transform infrared (ATR-FTIR) spectrophotometer (INVENIO-S, Bruker Corporation, Karlsruhe, Germany) was used to record the ATR-FTIR spectra of SA, TPA, and biocomposites. The spectrophotometer had a frequency range of 500–4000 cm^−1^, a resolution of 4 cm^−1^, and an accumulation of 32 scans.

A thermogravimetric analyzer (TGA, SDT Q600, TA-Instruments, New Castle, DE, USA) was used to assess the thermal decomposition characteristics of SA, TPA, and biocomposites. A nitrogen flow of 100 mL/min was used to heat 5–10 mg of each sample from 50 to 800 °C at a rate of 20 °C/min for analysis.

An X-ray diffractometer (XRD, D8 Advance, Bruker Corporation, Karlsruhe, Germany) with Cu-Kα radiation set to 40 kV and 40 mA was used to determine the crystalline structures of the SA and TPA. The scanning range was from 5° to 30°, and the step width was 0.02°.

A differential scanning calorimeter (DSC, Pyris Diamond, PerkinElmer, Waltham, MA, USA) operating under a nitrogen flow was used to examine the thermal transition characteristics of biocomposites. The samples were rapidly quenched to 0 °C after being heated at 200 °C for three minutes to erase their thermal history. The samples were then heated at a rate of 10 °C/min from 0 to 200 °C. The following formula was used to determine the degree of crystallinity (*X_c_*) for PLA crystallites based on the enthalpies of melting (Δ*H_m_*) and cold crystallization (Δ*H_cc_*).*X_c_* (%) = [(Δ*H_m_* − Δ*H_cc_*)/(93.6 × *W_PLA_*)] × 100(1)
where 93.6 J/g is a Δ*H_m_* of 100%*X_c_* PLA [29]. *W_PLA_* is the weight fraction of PLA.

A scanning electron microscope (SEM, TM4000Plus, HITACHI, Tokyo, Japan) operating at a 15 kV acceleration voltage was used to examine the morphology of the cryo-fractured surfaces of biocomposites. Liquid nitrogen was used to fracture the film samples. Prior to SEM scanning, a thin layer of gold was sputter-coated onto the film samples.

A universal testing machine (LY-1066B, Dongguan Liyi Environmental Technology Co., Ltd., Dongguan, China) with a 100 kg load cell was used to measure the tensile characteristics of film samples at 25 °C with an initial gauge length of 50 mm. A cross-head speed of 50 mm/min was used. Film samples were cut into 100 mm length and 10 mm width for the tensile test. The average of five measurements was used to calculate the tensile properties.

A contact angle analyzer (OCA11, DataPhysics Instruments, Filderstadt, Germany) was used to measure the water contact angle of biocomposites using the sessile drop method. Fifteen seconds after the deionized water (2.5 µL) was dropped from the left and right sides of the water droplet, the contact angles on the film surface were measured and then averaged. Five calculations were made and averaged for every film sample.

The soil burial degradation tests of 20 × 20 mm film samples were conducted in soil according to the literature [30]. After being dried in a convection oven at 105 °C for 24 h, the film samples were weighed (*W_i_*). The film sample was subsequently buried 5.0 cm below the soil’s surface in a 1.0 mm nylon mesh. Every other day, the soil was watered. The soil’s parameters were maintained at 25–30 °C, a pH of 6.0–7.0, and a moisture content of 50–60%. At various intervals, representative film samples were taken from the soil and washed with distilled water. They were then weighed after being dried in a convection oven at 105 °C for 24 h (*W_t_*). Every measurement was made three times. The following equation was used to calculate the weight loss during the soil burial of the film samples.Weight loss in soil burial (%) = [(*W_i_* × *W_t_*)/*W_i_*] × 100(2)

## 3. Results and Discussion

### 3.1. Characterization of TPA

Figure 2 shows the ATR-FTIR spectra of SA and TPA. The ATR-FTIR spectrum of SA in Figure 2a shows two broad absorption bands of C-H bonds at 2920 cm^−1^ and stretching hydroxyl (-OH) groups in the range of 3000–3700 cm^−1^. Symmetric and asymmetric stretching absorption bands of carboxylate (-COO-) groups were demonstrated at 1419 cm^−1^ and 1615 cm^−1^, respectively [31,32], and the stretching absorption bands of C-O-C bonds were shown at 1075 cm^−1^ and 1025 cm^−1^ [27]. In Figure 2b, the ATR-FTIR spectrum of TPA appears similar to that of SA; adding glycerol did not change the chemical structure of alginate. However, the hydroxyl groups in glycerol made the absorption bands of -OH groups between 3000 and 3700 cm^−1^ more intense [27]. The added glycerol exhibited a shoulder band of the C-H groups at 2903 cm^−1^ [33].

Thermal decomposition properties of SA and TPA were determined from their thermogravimetric (TG) and derivative TG (DTG) thermograms, as shown in Figure 3. Two thermal decomposition stages were identified for SA from Figure 3a: the range of 50–150 °C, which was assigned to represent the evaporation of moisture, and the range of 200–500 °C, which was attributed to represent the thermal decomposition of cellulose [27]. The peaks in DTG thermograms in Figure 3b are assigned as the maximum weight loss temperature (*T_max_*) of the samples. The SA showed two *T_max_* peaks at 89 °C and 253 °C due to the evaporation of moisture and the thermal decomposition of cellulose, respectively [27]. The DTG thermogram of TPA has three *T_max_* peaks, indicating that there are three thermal decomposition stages for TPA: the 50–150 °C range was assigned to represent moisture evaporation, the 180–210 °C range was assigned to represent glycerol evaporation, and the 210–500 °C range was assigned to represent cellulose thermal decomposition [31,34].

We found the *T_max_* peak for TPA at 225 °C, suggesting that the cellulose component of TPA decomposed thermally at a lower temperature than SA. This finding may be explained by the plasticization of glycerol molecules between SA chains, which weakens their intermolecular bonds. This means that glycerol probably induced the thermal decomposition of TPA at a lower temperature than SA [27].

Figure 4 shows XRD profiles for SA and TPA to study their crystalline structures. Two broad XRD peaks at 13.9° and 22.0° were seen in SA, as shown in Figure 4a, indicating that there are two separate amorphous regions: one with longer and one with shorter distances between closest neighboring alginate chains [27,28]. In Figure 4b, the XRD peak at 22.0° was stronger when the SA was plasticized with glycerol to create TPA, indicating that the glycerol made the second amorphous phase more noticeable.

### 3.2. Characterization of PLA/TPA Biocomposites

Chemical structures of the biocomposites were determined by ATR-FTIR spectra, as presented in Figure 5. The ATR-FTIR spectrum of pure PLA in Figure 5a shows an absorption band at 1749 cm^−1^, attributed to the stretching of ester-carbonyl (C=O) groups; at 1181 cm^−1^, 1128 cm^−1^, and 1081 cm^−1^, attributed to the C-O-C in ester groups; and at 2996 cm^−1^, 2946 cm^−1^, and 2880 cm^−1^, attributed to the stretching of C-H bonds [35,36]. In the ATR-FTIR spectra of biocomposites, the absorption bands of asymmetric stretching carboxylate (-COO-) groups at 1615 cm^−1^ exhibited an increasing intensity trend as the TPA content increased [31]. This study confirmed that the TPA content in the biocomposites increased with the feed TPA content.

The hydroxyl regions of the ATR-FTIR spectra of biocomposites are displayed in Figure 5b. It is clearly seen that the intensities of the -OH absorption bands steadily increased as the TPA content increased. In PLA/TPA biocomposites, the wavenumbers for the absorption bands of -OH groups decreased as the TPA content increased. This suggests that hydrogen bonds were formed between the hydroxyl groups of TPA and the carbonyl groups of PLA [34].

The thermal transition properties of biocomposites were determined from the DSC thermograms, as illustrated in Figure 6, and the DSC results are summarized in Table 1. The pure PLA exhibited a glass transition temperature (*T_g_*) of 58 °C. The *T_g_* values of biocomposites were lower than those of pure PLA and decreased with increasing TPA content. This phenomenon can be attributed to glycerol diffusing from TPA phases into PLA matrices, thereby enhancing the plasticizing effect within the PLA phases. It has been reported that the plasticizing effect of glycerol from thermoplastic starch (TPS) decreased the *T_g_* values of PLA/TPS biocomposites [14,37,38].

The cold crystallization temperature (*T_cc_*) of the biocomposites was found to be lower than that of pure PLA and shifted to a lower temperature as the TPA content increased, indicating that adding TPA improved the crystallization of PLA matrices [39,40,41]. The plasticization effect of diffused glycerol facilitates PLA chain rearrangement for crystallization [42]. The melting temperature (*T_m_*) of the biocomposites also shifted to a lower temperature with increasing TPA content. This alteration could be due to the glycerol plasticizer promoting the imperfect PLA crystals [14]. The degree of crystallinity (*X_c_*) of pure PLA was found to be 8.1%, which increased up to 9.6% when 5% and 10% TPA were added, showing that the TPA phase acted as a heterogeneous nucleating agent. However, the *X_c_* values of the biocomposites decreased again when the TPA content was more than 10%. This result might be because the TPA phases are enlarging, as shown by the SEM analysis below, which reduces their nucleating effectiveness.

The thermal decomposition properties of biocomposites were analyzed from the TG and DTG thermograms, as shown in Figure 7a and Figure 7b, respectively, and the TGA results are summarized in Table 2. The TG thermogram of pure PLA exhibited a single thermal decomposition stage in the range of 300–450 °C without char residue. It was found that the thermal decomposition range of the PLA matrix shifted downward to a low temperature when 5% TPA was incorporated, indicating that the addition of TPA decreased the thermal stability of the PLA matrix. The thermal decomposition range of biocomposites shifted significantly to a lower temperature range as the TPA content increased. As would be expected, the char residue at 800 °C of biocomposites also increased with an increase in TPA content. This is due to the char residue at 800 °C of TPA being about 20% [see Figure 3a].

From the DTG thermograms in Figure 7b, the *T_max_* peaks were identified and summarized in Table 2. The pure PLA exhibits a *T_max_* peak at 378 °C. The *T_max_* values of the biocomposites were lower than those of pure PLA, confirming that the incorporation of TPA leads to decreased thermal stability in the biocomposites compared to pure PLA. The *T_max_* peak of the biocomposites shifted to a lower temperature as TPA content increased, due to the migrated glycerol acting as a plasticizer, which weakens the intermolecular bonds between PLA chains and reduces the *T_max_* value. Similar results have been observed in the case of PLA/TPS biocomposites [43]. At around 220 °C, small *T_max_* peaks were observed, resulting from the TPA thermal decomposition. The intensity of these peaks increased as the TPA content increased, indicating poor phase compatibility between the PLA and the TPA, as demonstrated by the SEM analysis below.

The phase morphology of biocomposites was analyzed from SEM images of cryo-fractured surfaces, as shown in Figure 8. The analysis revealed flat fracture surfaces in the pure PLA, which indicates its brittleness. Biocomposites exhibit dispersed TPA particles within the PLA matrix, suggesting that they are immiscible blends. The interactions between the immiscible blend components could weaken [44]. Biocomposites containing 5% and 10% TPA show smaller particle sizes compared to those containing 20% and 30% TPA. As expected, incorporating a high amount of TPA led to the coalescence of the TPA phases [16]. Additionally, we observed gaps between the PLA matrix and the TPA particles, indicating their poor phase compatibility. This effect is due to the differing hydrophilicity of PLA and TPA, with TPA exhibiting a higher hydrophilicity. The results were further validated by measuring the water contact angle, as illustrated below. The size of the gaps increased significantly when the TPA contents increased up to 20% and 30%, as shown in Figure 8i,j, respectively. We also saw some cracks in the PLA matrix for the biocomposites with 20% and 30% TPA, showing that the PLA matrix became more brittle, as shown by the tensile test below.

The mechanical properties of biocomposites were investigated from tensile curves, as shown in Figure 9, and the results are summarized in Table 3. Pure PLA exhibited a maximum tensile strength of 63.2 MPa, an elongation at break of 3.4%, and a Young’s modulus of 1090 MPa. When TPA was added, it was discovered that the tensile properties of the PLA matrix decreased because the interface bond between the PLA and the TPA was weak, which made stress transfer less effective. Jozinovic et al. [16] and Akrami et al. [45] reported similar results, where the tensile properties decreased in the PLA matrix with the addition of TPS. The most important factor that contributes to the strength of two-phase composite polymers is effective stress transfer between the polymer matrix and the dispersed particles. The stress transfer at the particle/polymer interface was inefficient for weakly bound particles. Debonding occurs due to the poor interfacial adhesion between the dispersed particles and the polymer matrix. Thus, the dispersed particles were unable to carry any load, and as particle content increased, the composite strength decreased [46]. Some compatibilizers, such as maleic anhydride [47,48], formamide [49], and citric acid (CA) [50], have been used to improve the phase compatibility of the PLA/TPS biocomposites, which increases the tensile strength and elongation at break of PLA/TPS biocomposites by reducing the size of the TPS phases and increasing the interfacial adhesion between the components [50]. Therefore, we can use these compatibilizers to enhance the phase compatibility and mechanical properties of PLA/TPA biocomposites.

The biocomposites with 5%, 10%, 20%, and 30% TPA exhibited reductions in maximum tensile strengths of 21%, 34%, 54%, and 66%, respectively, compared to pure PLA. The biocomposites with 5%, 10%, 20%, and 30% TPA showed a drop in elongation at break of 2.9%, 11.8%, 44.1%, and 52.9%, respectively, when compared to pure PLA. The tensile results show that adding more than 10% TPA leads to a dramatic reduction in the mechanical properties, which correlates with the observed larger gaps between the PLA matrix and the dispersed TPA particles in their SEM images, as shown in Figure 8d,i and Figure 8e,j for the 20% and 30% TPA-loaded biocomposites, respectively. The larger gaps between the components were defect points that significantly reduced the mechanical properties.

The hydrophilicity of biocomposites can be determined by their water contact angles, as shown in Figure 10. It was observed that pure PLA exhibited the highest water contact angle of 81.9°, indicating its high hydrophobicity [51]. The water contact angles decreased continuously with increasing TPA content, suggesting that the biocomposites became more hydrophilic as TPA content increased. The FTIR results support this trend by demonstrating an increase in the intensity of -OH absorption bands. We attribute this change to TPA’s greater hydrophilicity compared to pure PLA.

Figure 11 illustrates the film characteristics made from pure PLA and its biocomposites. The pure PLA film is the clearest, while the PLA/TPA biocomposite films appear more hazy due to the dispersion of TPA phases. Additionally, the films become a darker brown as the TPA content increases; however, the letters beneath the films remain clearly legible for all the biocomposite films. The addition of TPA affects the optical transparency and color of the films. These biocomposites may be suitable for applications where visual clarity is important, yet enhanced material properties are desired.

A degradation test for soil burial was carried out for 6 months. Figure 12 illustrates the characteristics of the soil burial film samples. It was noted that the pure PLA showed no significant changes and stayed clear, suggesting that very little water entered the film matrix, which aligns with previous results [30]. After being buried in soil for a month, we found that all the biocomposites had become opaque, showing that water had diffused into the PLA matrix [16]. We also noticed some empty voids (or pores) in the films, which means they were degraded, probably because the TPA parts were being degraded. Soil conditions, including temperature, pH, and the presence of microorganisms, are critical factors influencing the soil burial degradation of TPA, thereby affecting the physical and chemical properties [16,52]. Reports indicate that the degradation of both PLA [53] and alginate [34] through soil burial involves two degradation steps. First, they absorb water from the soil, which leads to degradation through hydrolysis. The products of their hydrolysis are then degraded by soil microorganisms. The results of measuring the water contact angle mentioned above confirmed the high hydrophobicity of pure PLA, which accounts for this observation. This hydrophobic nature prevented water from permeating the material, thereby maintaining its clarity and structural integrity over time. Consequently, the film’s durability in soil conditions aligns with previous studies on PLA’s high resistance to moisture [53,54]. The TPA phase, due to its higher hydrophilicity, undergoes faster hydrolysis and degradation by soil microorganisms compared to PLA.

Furthermore, both the number and size of voids within the films increased with prolonged soil burial time and higher TPA content. Over time, the exposure of the biocomposites to environmental conditions intensified the degradation processes, leading to more significant structural changes. The correlation between void formation and TPA content underscores the impact of material composition on the rate of degradation during soil burial.

The ATR-FTIR spectra of the PLA/30% TPA biocomposites for different soil burial times prove this hypothesis, as shown in Figure 13. The intensities of the -OH absorption bands between 3000 and 3700 cm^−1^ of TPA components clearly decreased considerably as the soil burial time increased. This indicates that the TPA phase saw the majority of the soil burial degradation, and associated peaks vanished once this phase was degraded. PLA/TPS biocomposites produce results comparable to those described [45].

The soil burial degradation of biocomposites was also determined by their weight loss in soil burial, as shown in Figure 14. It was found that pure PLA showed no weight loss after 6 months in the soil, indicating that it degraded very slowly, which suggests that it may take a long time to decompose single-use PLA packaging in the soil. Our findings show that biocomposites lost more weight in the soil than pure PLA, meaning that biocomposites’ soil burial degraded faster than pure PLA. The weight loss during the soil burial of biocomposites increased with a longer soil burial time and a higher TPA content. Thus, adding TPA effectively accelerated the rate of degradation during soil burial of the PLA-based biocomposites. The presence of void structures in the soil burial biocomposites made it easier for moisture and microorganisms to enter the PLA matrix, which accelerated the rate of PLA’s soil burial degradation.

## 4. Conclusions

Thermoplastic alginate (TPA) was successfully prepared using glycerol as a plasticizer. The chemical structure of TPA is similar to that of sodium alginate (SA), as indicated by FTIR results. The TPA contained more -OH groups compared to SA. However, it exhibits lower thermal stability than SA, as demonstrated by TGA results. Furthermore, the microstructure of TPA is more homogeneous in its amorphous regions than SA, as shown in the XRD results.

PLA/TPA biocomposites were successfully prepared using the melt blending technique. The FTIR results indicated that PLA and TPA were interacting, which was indicated by the -OH bands shifting to a lower wavenumber. The DSC results showed that the PLA matrix used the TPA phases to improve the crystallization, which was indicated by the increasing *X_c_* values. However, the nucleating effectiveness of TPA diminished when its content exceeded 10%. Additionally, the incorporation of TPA reduced the thermal stability of the PLA matrix, as shown in the TGA results. The *T_max_* peaks of the biocomposites shifted to lower temperatures as the TPA content increased. The SEM analysis showed that the PLA and TPA components had poor phase compatibility because they have different hydrophilicities. The gaps between the PLA and the TPA phases were then observed. The size of TPA phases tended to increase significantly when the TPA content exceeded 10 wt%. The ultimate tensile strength, elongation at break, and Young’s modulus of the biocomposites from the tensile test decreased as the TPA content increased. These tensile properties dramatically decreased when the TPA content was higher than 10 wt%. The hydrophilicity of the biocomposites increased steadily as the TPA content increased, as shown by the results of the water contact angle. For the 6-month period of soil burial degradation, the pure PLA did not change significantly. In contrast, all the biocomposites buried in soil for 6 months showed both bulk and surface erosion, with noticeable empty voids and changes in the shape of the sample films. All the biocomposites exhibited a large weight loss value during the first month of soil burial before increasing slowly. The weight loss during the soil burial of biocomposites increased steadily as the TPA content increased. In summary, by changing the TPA content, we can find a satisfactory balance between the mechanical properties and the soil burial degradation rate of these biocomposites for certain single-use packaging needs.

## Figures and Tables

**Figure 1 polymers-17-01338-f001:**
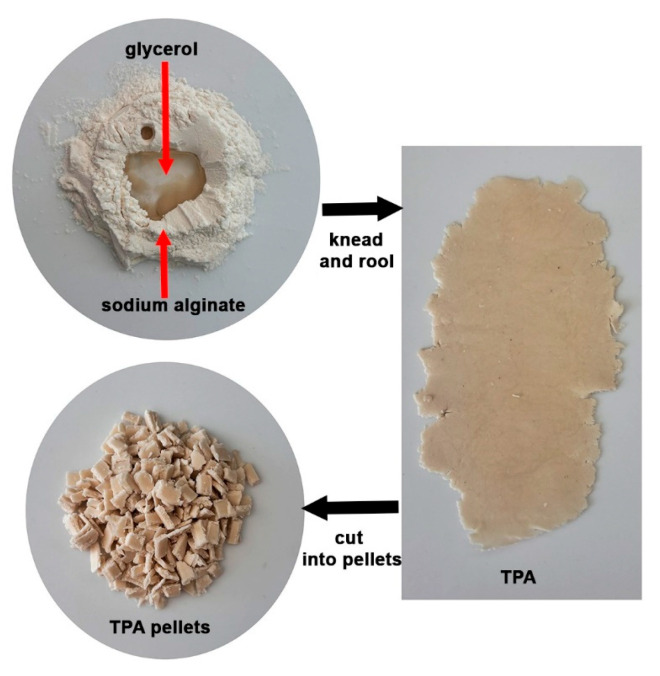
Preparation process of TPA pellets.

**Figure 2 polymers-17-01338-f002:**
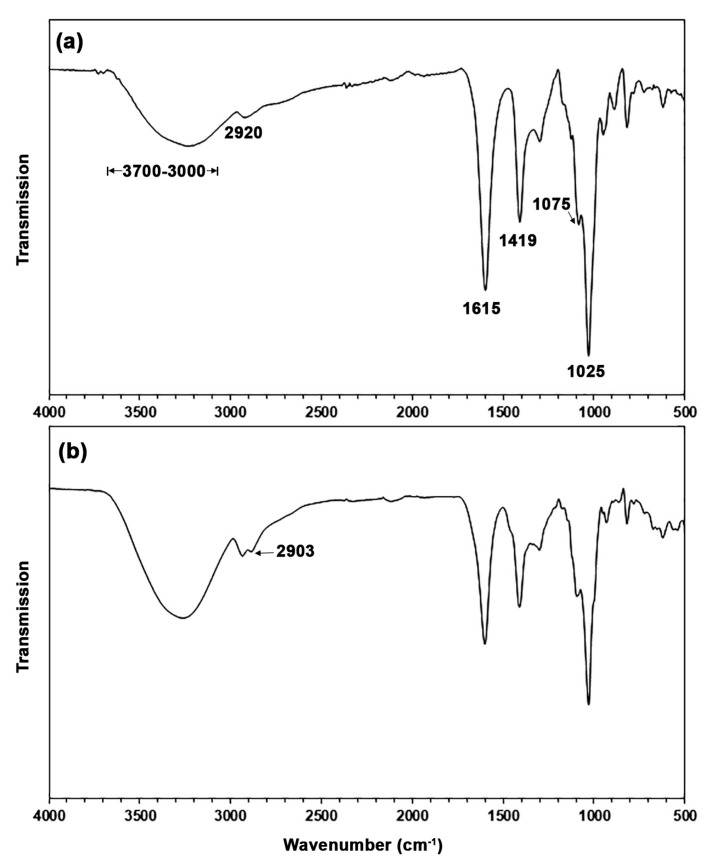
ATR-FTIR spectra of (**a**) SA and (**b**) TPA.

**Figure 3 polymers-17-01338-f003:**
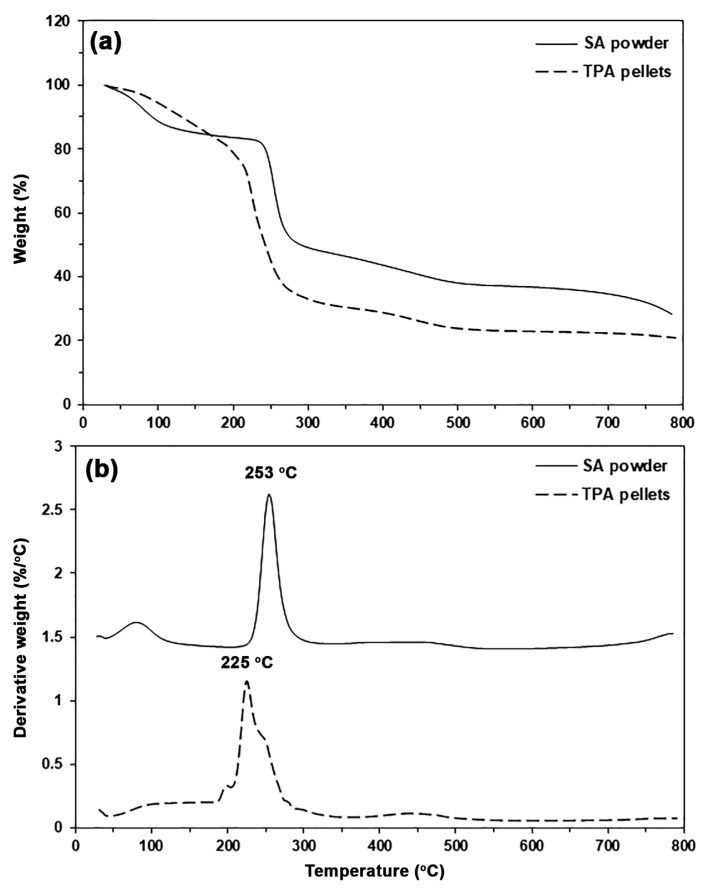
(**a**) TG and (**b**) DTG thermograms of SA and TPA.

**Figure 4 polymers-17-01338-f004:**
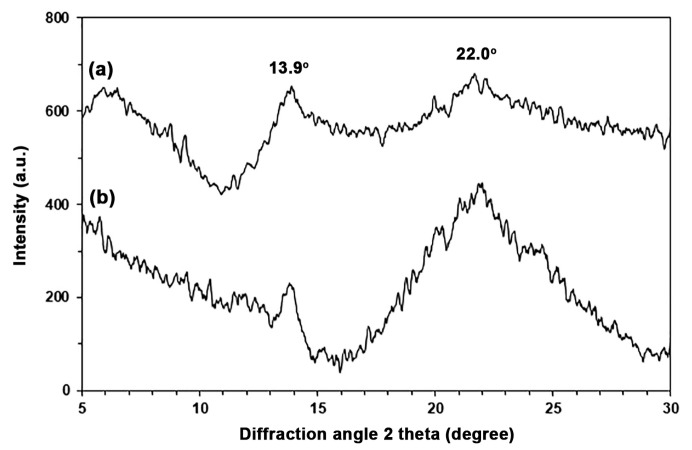
XRD profiles of (a) SA and (b) TPA.

**Figure 5 polymers-17-01338-f005:**
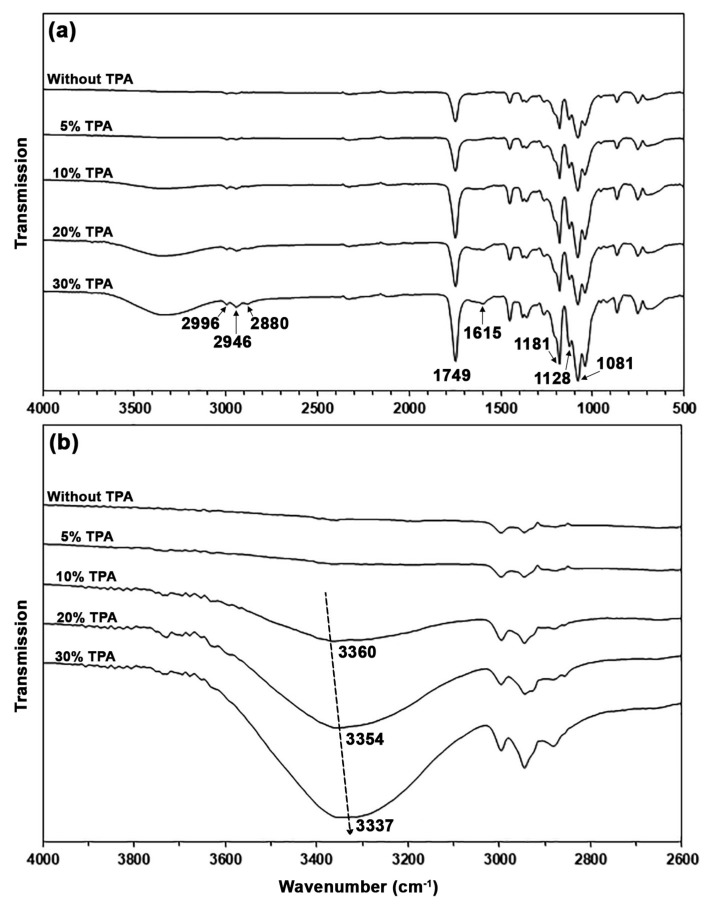
(**a**) ATR-FTIR spectra and (**b**) the expanded hydroxyl regions of PLA/TPA biocomposites with various TPA contents.

**Figure 6 polymers-17-01338-f006:**
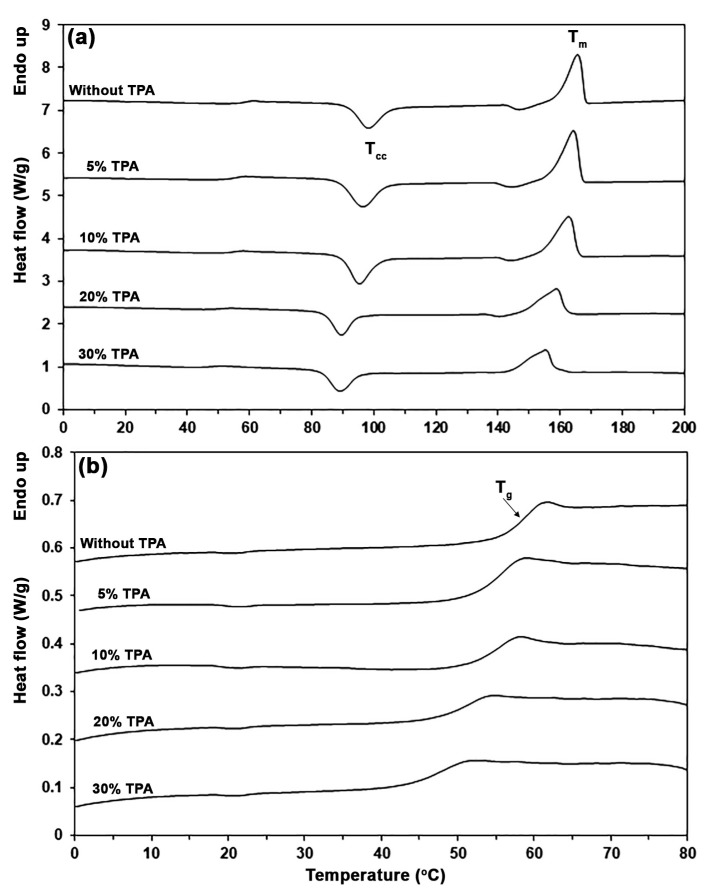
(**a**) DSC thermograms and (**b**) expanded *T_g_* regions of PLA/TPA biocomposites with various TPA contents.

**Figure 7 polymers-17-01338-f007:**
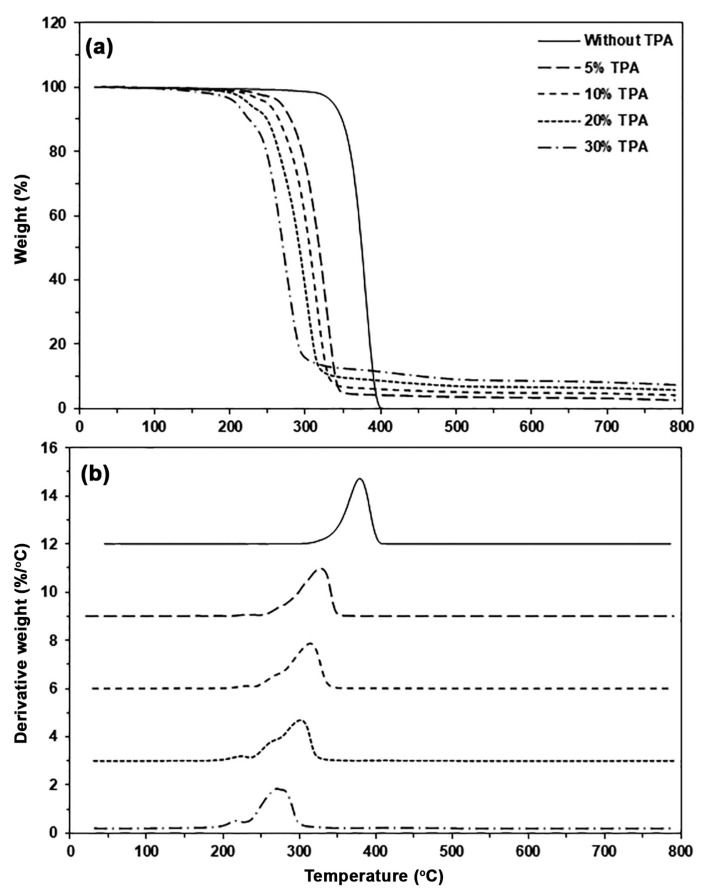
(**a**) TG and (**b**) DTG thermograms of PLA/TPA biocomposites with various TPA contents.

**Figure 8 polymers-17-01338-f008:**
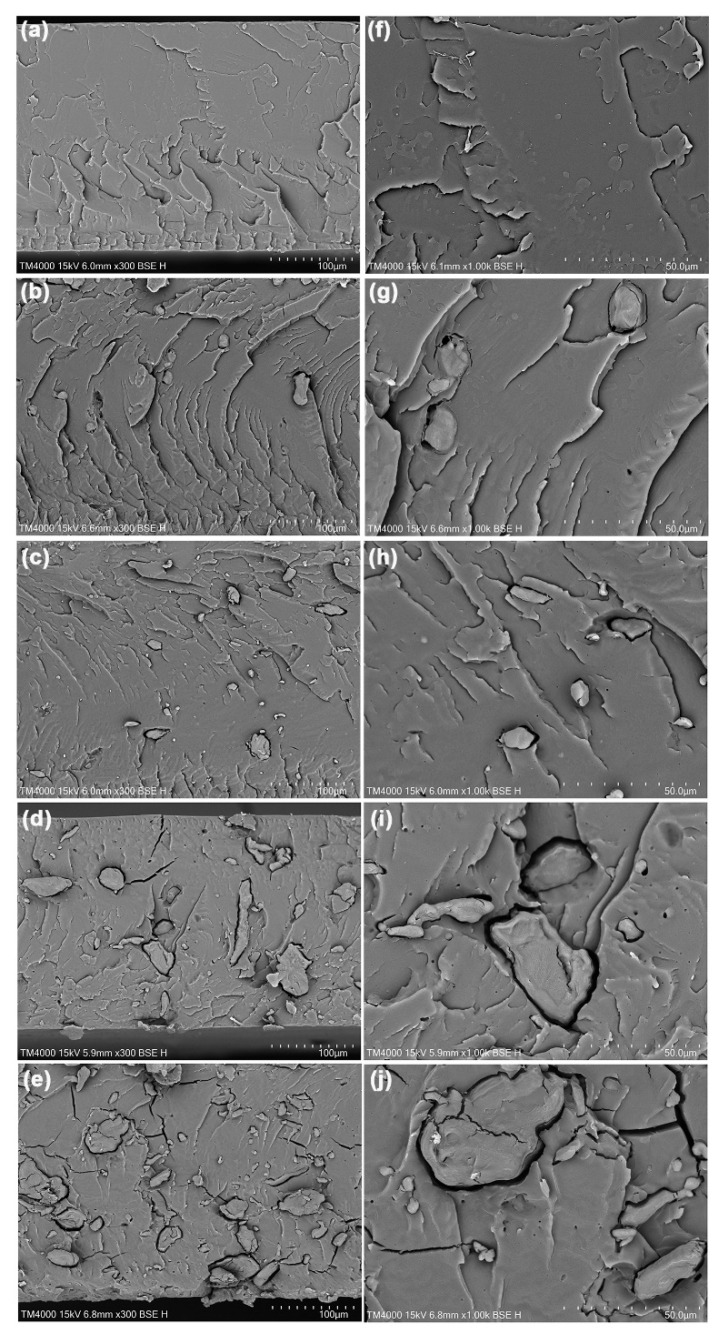
SEM images of cryo-fractured surfaces of (**a**,**f**) pure PLA and PLA/TPA biocomposites with TPA contents of (**b**,**g**) 5 wt%, (**c**,**h**) 10 wt%, (**d**,**i**) 20 wt%, and (**e**,**j**) 30 wt% for magnifications of (**a**–**e**) 300× and (**f**–**j**) 1000×.

**Figure 9 polymers-17-01338-f009:**
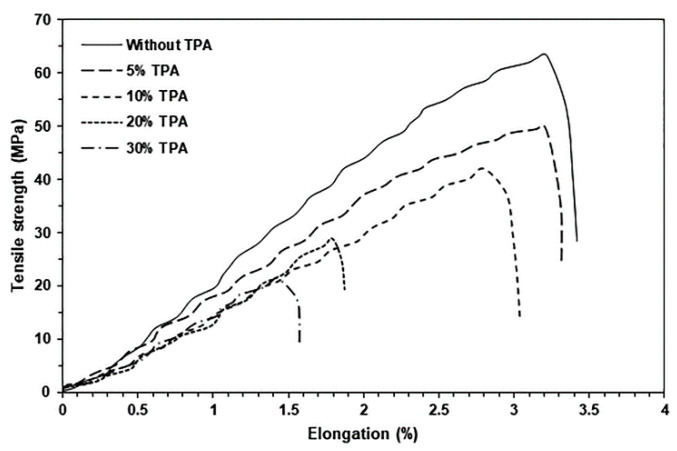
Selected tensile curves of PLA/TPA biocomposites with various TPA contents.

**Figure 10 polymers-17-01338-f010:**
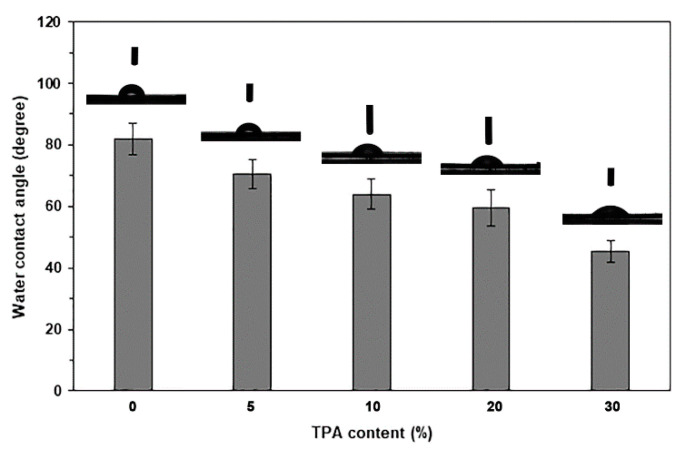
Water contact angle results of PLA/TPA biocomposites with various TPA contents. Values are given as mean ± SD (*n* = 5).

**Figure 11 polymers-17-01338-f011:**
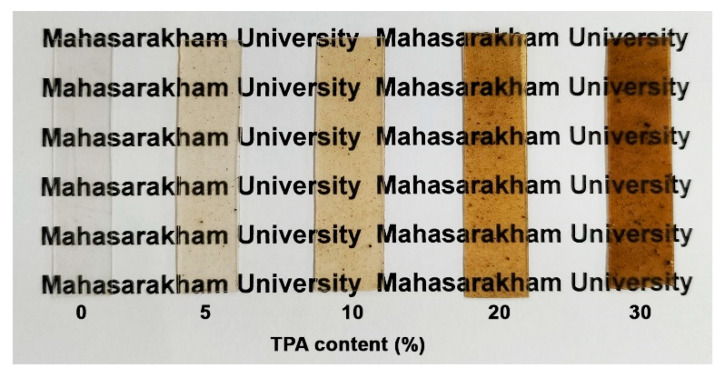
Pure PLA and PLA/TPA biocomposites with various TPA contents.

**Figure 12 polymers-17-01338-f012:**
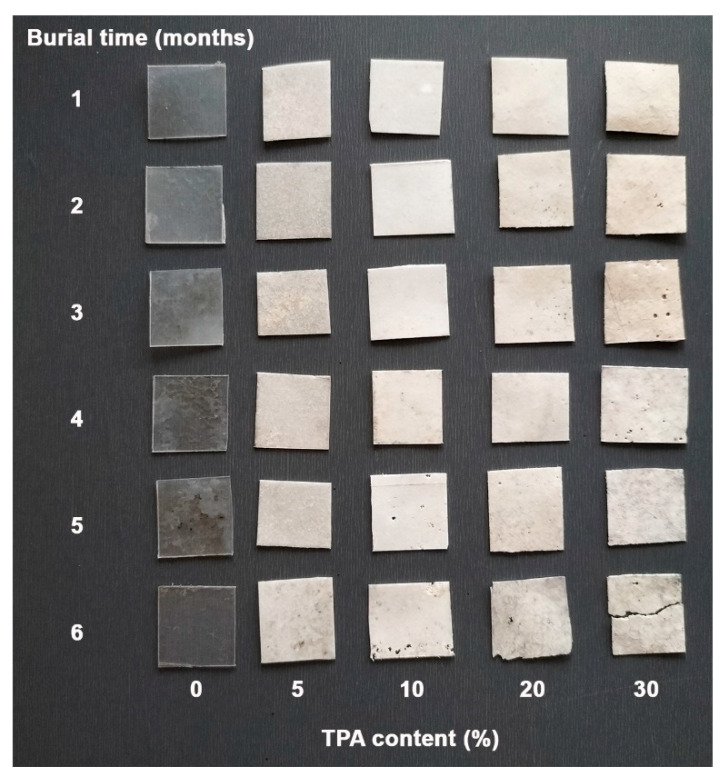
Visual changes after soil burial of pure PLA and PLA/TPA biocomposites with various TPA contents for various soil burial degradation times.

**Figure 13 polymers-17-01338-f013:**
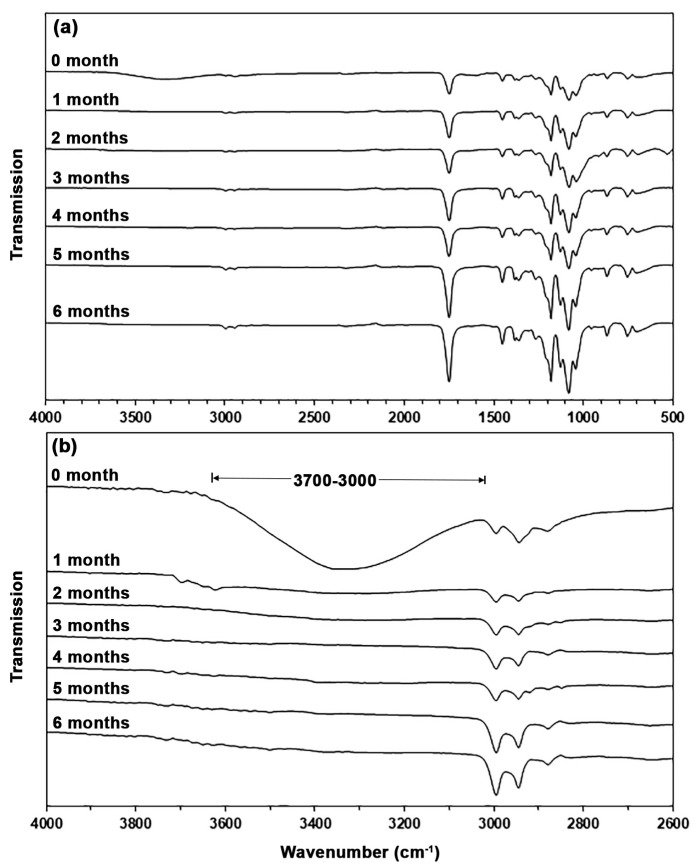
(**a**) ATR-FTIR spectra and (**b**) expanded hydroxyl regions of PLA/30% TPA biocomposites for various soil burial degradation times.

**Figure 14 polymers-17-01338-f014:**
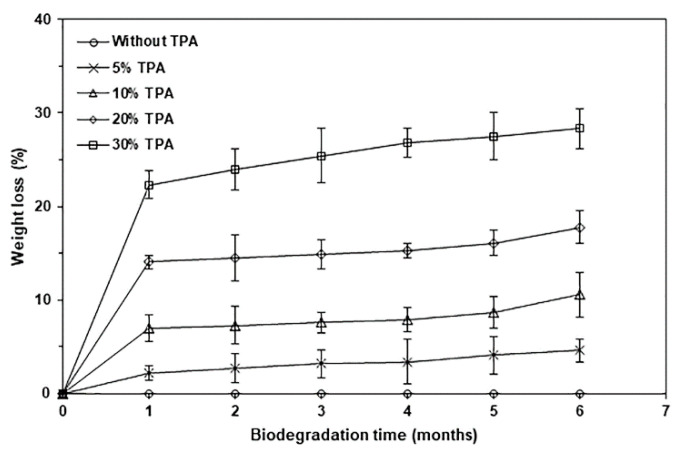
Weight losses in soil burial of PLA/TPA biocomposites with various TPA contents for various soil burial degradation times. Values are given as mean ± SD (*n* = 3).

**Table 1 polymers-17-01338-t001:** Thermal transition properties of PLA/TPA biocomposites.

TPA Content (wt%)	*T_g_* (°C)	*T_cc_* (°C)	Δ*H_cc_* (J/g)	*T_m_* (°C)	Δ*H_m_* (J/g)	*X_c_* (%)
-	58	98	32.8	166	40.4	8.1
5	54	97	32.1	164	40.6	9.6
10	54	95	32.0	163	40.1	9.6
20	50	90	22.9	159	29.7	9.1
30	46	89	21.6	155	26.9	8.1

**Table 2 polymers-17-01338-t002:** Thermal decomposition properties of PLA/TPA biocomposites.

TPA Content (wt%)	Char Residue at 800 °C (%)	*T_max_* (°C)
-	-	378
5	2.65	328
10	4.21	315
20	5.84	302
30	7.43	270

**Table 3 polymers-17-01338-t003:** Tensile properties of PLA/TPA biocomposites.

TPA Content (wt%)	Maximum Tensile Strength (MPa) *	Elongation at Break (%) *	Young’s Modulus (MPa) *
-	63.2 ± 3.8	3.4 ± 1.2	1090 ± 28
5	49.9 ± 4.4	3.3 ± 0.8	801 ± 37
10	41.8 ± 3.7	3.0 ± 0.9	692 ± 47
20	28.9 ± 3.5	1.9 ± 0.7	394 ± 28
30	21.3 ± 2.4	1.6 ± 0.4	180 ± 19

* Values are given as mean ± SD (*n* = 5).

## Data Availability

Data are contained within the article.

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
