# Peer review of "Properties and Biodegradation of Poly(lactic Acid)/Thermoplastic Alginate Biocomposites Prepared via a Melt Blending Technique"

_polymers, 2025, doi:10.3390/polym17101338_

Round 1

Reviewer 1 Report

Comments and Suggestions for Authors

In this manuscript, the author provides a detailed characterization of the properties of polylactic acid/thermoplastic alginate nanocomposites prepared by melt blending method. The influence of TPA content on the properties of composite materials was described in detail through some characterization and testing methods, which has research significance. But before publication, some modifications were made.

  1. It is suggested that the author indicate the positions of the functional groups mentioned in the FTIR images involved in the article and modify the vertical axis in the image to Transmission.
  2. The author should pay attention to the format of the horizontal and vertical axis names in the image, whether the first letter should be capitalized uniformly, whether the font size in the entire image is consistent, and maintain consistency in the image format throughout the text.
  3. Please ask the author to supplement the FTIR section of the characterization explanation of TPA in 3.1 and provide a detailed explanation of the viewpoint to be validated.
  4. In 3.2, please ask the author to double check if the calculated crystallinity value is correct.
  5. Suggest the author to refine the conclusion sections into more detailed points.

Comments on the Quality of English Language

The English could be improved to more clearly express the research

Author Response

Reviewer # 1_Round 1

Manuscript ID: polymers-3631815

Title: Properties and Biodegradation of Poly(lactic acid)/Thermoplastic Alginate Biocomposites Prepared via a Melt Blending Technique

Authors: Yodthong Baimark*, Kansiri Pakkethati and Prasong Srihanam

Reviewer 1:

In this manuscript, the author provides a detailed characterization of the properties of polylactic acid/thermoplastic alginate nanocomposites prepared by melt blending method. The influence of TPA content on the properties of composite materials was described in detail through some characterization and testing methods, which has research significance. But before publication, some modifications were made.

Authors:

Thank you for your letter about our manuscript entitled “Properties of Poly(lactic acid)/Thermoplastic Alginate Biocomposites Prepared via a Melt Blending Technique”. Those comments are very helpful for revising and improving our manuscript, as well as the important guiding significance to our research. We have carefully revised the manuscript according to your suggestions. The revised portions were marked in red. We also responded point by point to each comment as listed below, along with a clear indication of the location of the revision. We sincerely hope this manuscript will finally be acceptable to be published in Polymers. The responses to reviewer 1’s comments are as follows:

Comment 1: It is suggested that the author indicate the positions of the functional groups mentioned in the FTIR images involved in the article and modify the vertical axis in the image to Transmission.

Authors: The positions of the functional groups have been mentioned in the FTIR images on Figs. 1 (page 5) and 4 (page 7) of the revised manuscript. All the vertical axes in the FTIR images have been changed to "Transmission" in Figs. 1 (page 5), 4 (page 7), and 12 (page 15) of the revised manuscript as follows:

Figure 1. ATR-FTIR spectra of (a) SA and (b) TPA.

Figure 4. (a) ATR-FTIR spectra and (b) its expanded hydroxyl regions of PLA/TPA biocomposites with various TPA contents.

Figure 12. (a) ATR-FTIR spectra and (b) its expanded hydroxyl regions of PLA/30% TPA biocomposites for various soil burial degradation times.

Comment 2: The author should pay attention to the format of the horizontal and vertical axis names in the image, whether the first letter should be capitalized uniformly, whether the font size in the entire image is consistent, and maintain consistency in the image format throughout the text.

Authors: Thank you for your suggestions. The format and font size of all the figures have been revised.

Comment 3: Please ask the author to supplement the FTIR section of the characterization explanation of TPA in 3.1 and provide a detailed explanation of the viewpoint to be validated.

Authors: The revised manuscript adds more explanation from the FTIR results on page 6, lines 173-177 as follows:

In Figure 1(b), the ATR-FTIR spectrum of TPA appeared similar to that of SA; adding glycerol did not change the chemical structure of alginate. However, the hydroxyl groups in glycerol made the absorption bands of -OH groups between 3000 and 3700 cm⁻¹ more intense [27]. The added glycerol exhibited a shoulder band of the C-H groups at 2903 cm⁻¹ [33].

Comment 4: In 3.2, please ask the author to double check if the calculated crystallinity value is correct.

Authors: The calculated crystallinity values ​​in Table 1 (page 8) have been rechecked. These values ​​were found to be correct.

Comment 5: Suggest the author to refine the conclusion sections into more detailed points.

Authors: We have revised the conclusion section. The revised manuscript now includes more detail on pages 15 and 16 as follows:

  1. Conclusions

Thermoplastic alginate (TPA) was successfully prepared using glycerol as a plasticizer. The chemical structures of TPA are similar to that of sodium alginate (SA), as indicated by FTIR results. The TPA contained more -OH groups compared to SA. However, it exhibits lower thermal stability than SA, as demonstrated by TGA results. Furthermore, the microstructure of TPA is more homogeneous in its amorphous regions than that of SA, as shown in XRD results.

PLA/TPA biocomposites were successfully prepared using the melt blending technique. The FTIR results indicated that PLA and TPA were interacting, which was indicated by the -OH bands shifting to a lower wavenumber. The DSC results showed that the PLA matrix used the TPA phases to improve the crystallization, which was indicated by the increasing Xc values. However, the nucleating effectiveness of TPA diminished when its content exceeded 10%. Additionally, the incorporation of TPA reduced the thermal stability of the PLA matrix, as shown in TGA results. The Tmax peaks of the biocomposites shifted to lower temperatures as TPA content increased. The SEM analysis showed that the PLA and TPA components were poor phase compatibility because they have different hydrophilicity. The gaps between the PLA and the TPA phases were then observed. The size of TPA phases tended to increase significantly when the TPA content exceeded 10 wt%. The ultimate tensile strength, elongation at break, and Young’s modulus of the biocomposites from the tensile test decreased as the TPA content increased. These tensile properties dramatically decreased when the TPA content was higher than 10 wt%. The hydrophilicity of the biocomposites increased steadily as the TPA content increased, as shown by the results of water contact angle. For the 6-month period of soil burial degradation, the pure PLA did not change significantly. In contrast, all the biocomposites buried in soil for 6 months showed both bulk and surface erosion, with the noticeable empty voids and changes in the shape of the sample films. All the biocomposites exhibited a large weight loss value during the first month of soil burial before increasing slowly. The weight loss in soil burial of biocomposites increased steadily as the TPA content increased. In summary, by changing the TPA content, we can find a satisfactory balance between the mechanical properties and the soil burial degradation rate of these biocomposites for certain single-use packaging needs.

Comment 6: The English could be improved to more clearly express the research

Authors: English language experts have improved the English language to be clearer.

Concluding Remarks

We hope that our responses answer the reviewers’ comments to their satisfaction and that the revisions that have been made to the paper enhance its clarity for the benefit of the reader.

Yours Faithfully,

The Authors

Reviewer 2 Report

Comments and Suggestions for Authors

This manuscript presents a study on PLA/thermoplastic alginate (TPA) biocomposites, and gives interesting insights into the development of biodegradable materials. The experimental structure and analysis of the materials are appropriate for this journal and the results provide new insights in the field. 

However, several improvements are recommended to the authors to improve the manuscript's clarity, quality and scientific depth.

  1. The authors report the significant decrease of mechanical properties with increasing the TPA content, but a discussion on possible improvements of this phenomenon should be included (section 3.2). A suggested operation threshold could also be noted by the authors.
  2. Similarly, the SEM images show the phase incompatibility, but a more detailed discussion this observation should be included. A suggestion is to include a paragraph explaining possible interfacial adhesion mechanisms and also how these affect the reduction in mechanical strength. 
  3. The soil burial degradation test is a useful addition to this work, nevertheless, a molecular or mechanistic discussion of how the degradation takes place in this material is highly recommended. Is there a microbial attack or hydrolysis that initiate or progress the degradation faster?
  4. More detailed captions are suggested for all figures. Add the relevant information in the captions to make the standalone. Include information like conditions used, number of repeats, etc.
  5. Prior scientific contributions in the field are referenced, however, a deeper and more direct comparison to similar systems is suggested in the results/discussion section. 

Author Response

Reviewer # 2_Round 1

Manuscript ID: polymers-3631815

Title: Properties and Biodegradation of Poly(lactic acid)/Thermoplastic Alginate Biocomposites Prepared via a Melt Blending Technique

Authors: Yodthong Baimark*, Kansiri Pakkethati and Prasong Srihanam

Reviewer 2:

This manuscript presents a study on PLA/thermoplastic alginate (TPA) biocomposites, and gives interesting insights into the development of biodegradable materials. The experimental structure and analysis of the materials are appropriate for this journal and the results provide new insights in the field.

However, several improvements are recommended to the authors to improve the manuscript's clarity, quality and scientific depth.

Authors:

Thank you for your letter about our manuscript entitled “Properties of Poly(lactic acid)/Thermoplastic Alginate Biocomposites Prepared via a Melt Blending Technique”. Those comments are very helpful for revising and improving our manuscript, as well as the important guiding significance to our research. We have carefully revised the manuscript according to your suggestions. The revised portions were marked in red. We also responded point by point to each comment as listed below, along with a clear indication of the location of the revision. We sincerely hope this manuscript will finally be acceptable to be published in Polymers. The responses to reviewer 1’s comments are as follows:

Comment 1: The authors report the significant decrease of mechanical properties with increasing the TPA content, but a discussion on possible improvements of this phenomenon should be included (section 3.2). A suggested operation threshold could also be noted by the authors.

Authors: The revised manuscript's section 3.2 on page 12, lines 323-328, includes suggestions for improving this phenomenon as follows:

Some compatibilizers, such as maleic anhydride [47,48], formamide [49], and citric acid (CA) [50], have been used to improve the phase compatibility of the PLA/TPS biocomposites, which increases the tensile strength and elongation at break of PLA/TPS biocomposites by reducing the size of the TPS phases and increasing the interfacial adhesion between the components [50]. Therefore, we can use these compatibilizers to enhance the phase compatibility and mechanical properties of PLA/TPA biocomposites.

Comment 2: Similarly, the SEM images show the phase incompatibility, but a more detailed discussion this observation should be included. A suggestion is to include a paragraph explaining possible interfacial adhesion mechanisms and also how these affect the reduction in mechanical strength.

Authors: More detailed discussion on poor phase compatibility between the PLA and the TPA phases has been added on page 10, lines 289-303 of the revised manuscript. More explanation has been added on pages 11 and 12, lines 310-328 of the revised manuscript, about possible interfacial adhesion mechanisms and how these affect the reduction in mechanical strength as follows:

Page 10, lines 289-303

The phase morphology of biocomposites was analyzed from SEM images of cryo-fractured surfaces, as shown in Figure 7. The analysis revealed flat fracture surfaces in the pure PLA, which indicates its brittleness. Biocomposites exhibit dispersed TPA particles within the PLA matrix, suggesting that they are immiscible blends. The interactions between the immiscible blend components could weaken [44]. Biocomposites containing 5% and 10% TPA show smaller particle sizes compared to those containing 20% and 30% TPA. As expected, incorporating a high amount of TPA led to the coalescence of the TPA phases [16]. Additionally, we observed gaps between the PLA matrix and the TPA particles, indicating their poor phase compatibility. This effect is due to the differing hydrophilicity of PLA and TPA, with TPA exhibiting a higher hydrophilicity. The results were further validated by measuring the water contact angle, as illustrated below. The size of the gaps increased significantly when the TPA contents increased up to 20% and 30%, as shown in Figures 7(i) and 7(j), respectively. We also saw some cracks in the PLA matrix for the biocomposites with 20% and 30% TPA, showing that the PLA matrix became more brittle, as shown by the tensile test below.

Pages 11 and 12, lines 310-328

The mechanical properties of biocomposites were investigated from tensile curves, as shown in Figure 8, and the results are summarized in Table 3. Pure PLA exhibited a maximum tensile strength of 63.2 MPa, an elongation at break of 3.4%, and a Young’s modulus of 1090 MPa. When TPA was added, it was discovered that the tensile properties of the PLA matrix decreased because the interface bond between the PLA and the TPA was weak, which made stress transfer less effective. Jozinovic et al. [16] and Akrami et al. [45] reported similar results, where the tensile properties decreased in the PLA matrix with the addition of TPS. The most important factor that contributes to the strength of two-phase composite polymers is effective stress transfer between the polymer matrix and the dispersed particles. The stress transfer at the particle/polymer interface was inefficient for weakly bound particles. Debonding occurs due to the poor interfacial adhesion between the dispersed particles and the polymer matrix. Thus, the dispersed particles were unable to carry any load, and as particle content increased, the composite strength decreased [46]. Some compatibilizers, such as maleic anhydride [47,48], formamide [49], and citric acid (CA) [50], have been used to improve the phase compatibility of the PLA/TPS biocomposites, which increases the tensile strength and elongation at break of PLA/TPS biocomposites by reducing the size of the TPS phases and increasing the interfacial adhesion between the components [50]. Therefore, we can use these compatibilizers to enhance the phase compatibility and mechanical properties of PLA/TPA biocomposites.

Comment 3: The soil burial degradation test is a useful addition to this work, nevertheless, a molecular or mechanistic discussion of how the degradation takes place in this material is highly recommended. Is there a microbial attack or hydrolysis that initiate or progress the degradation faster?

Authors: The revised manuscript now includes a more thorough discussion of the soil burial degradation of PLA/TPA biocomposites on pages 13 and 14, specifically lines 378-387 as follows:

Reports indicate that the degradation of both PLA [53] and alginate [34] through soil burial involved two degradation steps. First, they absorbed water from the soil, which led to degradation through hydrolysis. The products of their hydrolysis were then degraded by soil microorganisms. The results of measuring the water contact angle mentioned above confirmed the high hydrophobicity of pure PLA, which accounts for this observation. This hydrophobic nature prevented water from permeating the material, thereby maintaining its clarity and structural integrity over time. Consequently, the film's durability in soil conditions aligns with previous studies on high PLA's resistance to moisture [53,54]. TPA phase, due to its higher hydrophilicity, undergoes faster hydrolysis and degradation by soil microorganisms compared to PLA.

Comment 4: More detailed captions are suggested for all figures. Add the relevant information in the captions to make the standalone. Include information like conditions used, number of repeats, etc.

Authors: We have revised the more detailed captions of Figs. 4-13. The number of repeats has been added to Figs. 9 and 13, as well as Table 3, of the revised manuscript.

Comment 5: Prior scientific contributions in the field are referenced, however, a deeper and more direct comparison to similar systems is suggested in the results/discussion section.

Authors: The revised manuscript includes improved references and comparisons with previous research that are more in-depth and straightforward.

Concluding Remarks

We hope that our responses answer the reviewers’ comments to their satisfaction and that the revisions that have been made to the paper enhance its clarity for the benefit of the reader.

Yours Faithfully,

The Authors
